# Understanding the Mechanical, Surface, and Color Behavior of Oral Bioactive Prosthetic Polymers under Biodegradation Processes

**DOI:** 10.3390/polym15112549

**Published:** 2023-05-31

**Authors:** Cristina B. Neves, Joana Costa, Jaime Portugal, Ana F. Bettencourt

**Affiliations:** 1Biomedical and Oral Sciences Research Unit (UICOB), Faculdade de Medicina Dentária, Universidade de Lisboa, 1600-277 Lisboa, Portugal; jaimeportugal@edu.ulisboa.pt; 2Egas Moniz Center for Interdisciplinar Research (CiiEM), Egas Moniz School of Health and Science, 2829-511 Almada, Portugal; jcosta@egasmoniz.edu.pt; 3Research Institute for Medicines (iMed.ULisboa), Faculdade de Farmácia, Universidade de Lisboa, 1649-003 Lisboa, Portugal

**Keywords:** drug delivery system, acrylic resins, color stability, denture reline, removable denture, antimicrobial

## Abstract

Changes in the properties of resin-based polymers exposed to the oral environment can emerge when chlorhexidine (CHX) is incorporated to develop bioactive systems for treating denture stomatitis. Three reline resins loaded with CHX were prepared: 2.5 wt% in Kooliner (K), 5 wt% in Ufi Gel Hard (UFI), and Probase Cold (PC). A total of 60 specimens were submitted to physical aging (1000 cycles of thermal fluctuations, 5–55 °C) or chemical aging (28 days of pH fluctuations in artificial saliva, 6 h at pH = 3, 18 h at pH = 7). Knoop microhardness (30 s, 98 mN), 3-point flexural strength (5 mm/min), and surface energy were tested. Color changes (ΔE) were determined using the CIELab system. Data were submitted to non-parametric tests (α = 0.05). After aging, bioactive K and UFI specimens were not different from the controls (resins without CHX) in mechanical and surface properties. Thermally aged CHX-loaded PC specimens showed decreased microhardness and flexural strength but not under adequate levels for function. The color change was observed in all CHX-loaded specimens that underwent chemical aging. The long-term use of CHX bioactive systems based on reline resins generally does not impair removable dentures’ proper mechanical and aesthetic functions.

## 1. Introduction

Tooth loss is associated with a decrease in the masticatory system’s integrity, which promotes functional, aesthetic, psychological, and quality-of-life negative consequences [1]. In the last decades, worldwide demographic and the average life expectancy increased the risk of tooth loss, especially among the elderly [2,3]. Despite the ultimate developments of fixed rehabilitation and implants, removable acrylic dentures are still valid treatment approaches for partial and total tooth loss, restoring chewing, phonetics, and aesthetics [2,4].

Polymethylmethacrylate (PMMA) is the most common resin-based polymer to manufacture these removable dentures [5].

Despite the advantages of acrylic resins in availability, good mechanical properties, and cost feasibility, they are sensitive to discoloration and represent excellent support for biofilm formation through the adhesion of microorganisms to their porous surface that may contribute to denture stomatitis [6,7].

Although the etiology of denture stomatitis is multifactorial, infection by *Candida* species, especially *Candida albicans*, is considered the primary etiologic factor [8,9,10].

The treatment of denture stomatitis includes the mechanical control of biofilm, the proper removal of the denture’s biofilm, and the pharmacological control using antimicrobial agents [11]. Systemic and topical antifungal treatment is advised, but recurrences are frequent, and one of the problems is that it requires daily compliance with the appropriate dosage to maintain its efficacy [8,12]. In addition, most of the antimicrobial agent is dissolved and removed from the oral cavity due to the renovation of saliva and the self-cleaning effect of the oral musculature. According to that, the efficacy of the topical agents is limited to short-term action [13,14].

For these reasons, drug delivery systems are being suggested based on incorporating an antimicrobial drug into acrylic resins [15,16]. This biomaterial system can be classified as bioactive material since when it enters in contact with the oral tissues, it starts to release the leachable antimicrobial drug, inducing a specific antimicrobial response at the material–tissue interface, such as the inhibition of biofilm formation and suppression of further microbial growth [17].

Compared to the conventional systemic therapeutic approach, these local delivery systems have advantages such as preserving therapeutic levels by continuous drug release at the infection site, minimal risk of systemic toxicity, and decreased need for patient compliance [18,19,20]. In this context, one antimicrobial with high interest to be incorporated into these systems is chlorhexidine (CHX), an antimicrobial agent widely prescribed as an antiseptic mouthwash in dentistry due to its broad-spectrum antimicrobial activity, including *C. albicans* [21].

Effectively, CHX has shown better results than other drugs such as fluconazole, both on releasing and microbiological tests [13,14,22]. Some studies have evaluated the CHX release when incorporated in resin-based prosthetic polymers and concluded that there is a high initial rate of elution from the material during the first 2–7 days, followed by a controlled, sustained elution process that continues throughout the 28-day test period [13,14]. A recent study showed promising results from a CHX-loaded reline acrylic resin used to enhance the fit of pre-existing ill-fitted dentures. The CHX release prevented microbiological development and did not negatively affect the resins’ mechanical, structural, and surface properties, showing a low cytotoxic profile [23]. Nevertheless, a major drawback in those studies refers to the testing methodologies. In fact, testing of dental devices is usually performed without the simulation of the oral environment’s influence on their properties, remaining some concerns about the long-term behavior of the devices after being in function and subjected to tissue fluids, blood, and saliva [24]. Oral biodegradation defines the changes in the devices’ chemical, physical, and mechanical properties due to environmental conditions that can ultimately compromise their function. Its processes can be produced by factors such as saliva constituents (e.g., water, enzymes), mechanical (e.g., chewing forces), physical (e.g., thermal variations), and chemical (e.g., pH variations) changes induced by daily eating, drinking, and breathing [25,26,27].

Concerning the evaluation of drug release profile from dental devices, most published studies use distilled water or artificial saliva at pH = 7 as the media solution to investigate drug liberation [12,14,18]; however, drug release has different behavior when submitted to acidic conditions and lower pH showed higher concentrations of leachable residual compounds [28]. Since the devices are exposed to endogenous and exogenous acids, it is essential to simulate pH variations that mimic the oral cavity environment. Moreover, pathological conditions such as denture stomatitis promote a lower pH environment and should be simulated when testing the properties of dental devices [29].

In the case of bioactive systems, these undergoing physical and chemical environmental changes also can affect the drug’s leachability and, consequently, the properties of the devices.

Another critical aspect is pointed out in some studies, which concluded that incorporating an antimicrobial drug into resin-based polymers produces color changes with clinical relevance when submitted to stainable liquids, such as coffee and wine [30,31,32]. In the case of CHX used as a mouthwash or gel, brownish staining of the teeth, mucosa, and biomaterials can cause unesthetic results [33]. However, studies on the color stability of CHX bioactive polymers are scarce, and that matter must be investigated.

Motivated by the gap in the literature concerning the oral biodegradation of dental devices, the main purpose of this work was to evaluate the mechanical, surface, and color of CHX-loaded polymeric resins after undergoing physical or chemical aging that mimics 1 month of exposition in the oral cavity. The null hypotheses were that the microhardness, flexural strength, surface energy, and color change of acrylic resins after each thermal or chemical aging process were not affected by the incorporation of CHX.

## 2. Materials and Methods

### 2.1. Resin-Based Polymers Preparation

Three auto-polymerized reline acrylic resins, with differences in their chemical composition and physical structure, were selected: two polyethylmethacrylate (PEMA)-based resins for direct curing on the oral cavity Kooliner (K) from GC America Inc., Alsip, IL, USA, and Ufi Gel Hard (UFI) from VOCO GmbH, Cuxhaven, Germany; and one PMMA-based resin for indirect curing in the laboratory Probase Cold (PC) from Ivoclar Vivadent AG Schaan, Liechtenstein. The CHX diacetate monohydrate (Panreac Applichem, Darmstadt, Germany) was selected to be incorporated in the reline acrylic resins (Table 1).

Two groups of specimens were established for each material: a control group with no drug incorporation and an experimental group with the incorporation of 2.5 wt% CHX (K) or 5 wt% CHX (UFI and PC). The selection of the CHX concentrations was based on the results of a previous study corresponding to the minimal concentration that promoted antimicrobial activity against *C. albicans* in each structurally distinct resin [23]. The correspondent CHX was mixed with the reline resin powder using a mortar and a pestle for homogenization. Then, the liquid monomer was added to the resulting powder, and the final mixture was placed into a specific mold maintained under compression at a time, temperature, and pressure recommended by the manufacturers (Table 1). The edges of each specimen were polished with 600-grit silicon carbide paper (Carbimet Paper Discs, Buehler Ltd., Lake Bluff, IL, USA) on a polisher with constant water cooling (DAP-U, Struers, Denmark) to remove irregularities.

### 2.2. Aging Processes

Control and experimental groups of specimens of each acrylic resin were then randomly divided to further submission to physical or chemical aging processes. The physical aging process included 1000 cycles of thermal fluctuations between 5 °C and 55 °C (20 s dwell time) with a transfer time of 5s in a thermocycling machine (Refri 200-E, Aralab, Cascais, Portugal) [34].

The chemical aging process included daily cycles of pH changes for 28 days. Each specimen was weighed on a precision scale (A&D Company, Limited, Tokyo, Japan) and immersed in artificial saliva with a ratio of 1 g/5 mL in individual graduated falcon tubes [27]. The specimens were stored at 37 °C (Memmert, Schwabach, Germany) with constant gentle shaking (300 rpm) for 28 days. The specimens were subjected to a protocol of chemical aging based on 6 h cycles in artificial saliva at pH = 3, interchanging with 18 h cycles in artificial saliva at pH = 7. Between each cycle, the samples were washed with distilled water and dried with absorbent paper.

### 2.3. Resin-Based Polymers Characterization

The mechanical properties of the bioactive polymers and controls after aging (thermal or chemical) were determined in rectangular-shaped specimens obtained by stainless-steel molds (64 × 10 × 3.3) mm [35] (*n* = 8). The Knoop microhardness test was performed (Duramin A/S, StruersDK 2750, Ballerup, Denmark) with a 98.12 mN (10.01 gf) load for 30 s. Twelve equidistant measurements were made in each specimen, and the mean value was considered the specimen’s Knoop microhardness (KHN—kgf mm^−2^). Immediately after the microhardness test, specimens were submitted to the three-point flexural test in a universal machine (Instron model 4502, Instron Ltd., Bucks, England) with 1 kN load cell at a crosshead speed of 5 mm/min and 50 mm between rods [35].

Another set of rectangular-shaped specimens (16 × 25 × 1) mm (*n* = 5) obtained from rectangular-shaped metallic strip molds were prepared to calculate the surface free energy after aging (thermal or chemical) with a tensiometer(K12, Kruss GmbH, Hamburg, Germany) using the Wilhelmy plate method by immersing plates into deionized water and 1,2-propanediol (Merck, Darmstadt, Germany) at a speed of 20 mm/s and 25 °C. Using the harmonic mean method, advancing contact angles estimated the specimens’ surface free energy (*γ*). Equations for surface free energy estimation were solved using the equation-handling software programsystem (K121 contact angle measuring version 2.05, Kruss GmbH, Hamburg, Germany) [36].

To determine the color change [37], groups of 12 × 6 mm cylindric specimens (*n* = 5) of each resin were prepared within 60 specimens. Color measurements were performed before and after aging processes using a spectrophotometer (Easyshade, VITA, Bad Säckingen, Germany) in a dark chamber. In each measurement, the L (lightness), C (chroma), and h (hue) values were registered. Since measurements of each specimen were performed in triplicates, an average value of L, C, and h was calculated per specimen. The CieLCh system C and h values were then converted to the CIELab system using the formulas C = (a^2^ + b^2^)^1/2^ and h = tang^−1^(b/a), and the L value was the same [31,32]. The overall color change (∆E) of each specimen was calculated through ∆E = [(∆L*)^2^ + (∆a*)^2^ + (∆b*)^2^]^1/2^. 

As National Institute of Standards and Technology recommends, the ∆E values were then converted to National Bureau of Standards (NBS) units using the formula NBS units = ∆E* × 0.92, which expresses the color differences from a clinical perspective as extremely slight change (NBS < 0.5), slight change (0.5 < NBS < 1.5), perceivable change (1.5 < NBS < 3), appreciable change (3 < NBS < 6), much appreciable (6 < NBS < 12), and change to another color (NBS > 12) [33].

### 2.4. Statistical Analysis

The sample size (*n*) was estimated with a power analysis to provide statistical significance (*a* = 0.05) at a power of 80%. Since normality and homogeneity of variance were not verified in scale variables (Shapiro–Wilk and Levene tests, *p* < 0.05), data were submitted to Mann–Whitney and Wilcoxon signed-rank tests (*α* = 0.05).

## 3. Results

Descriptive and statistical analysis of the microhardness, flexural strength, and surface free energy data was performed for each material (Table 2).

Microhardness medians variated between 7.0 ± 3.44 KHN in chemically aged CHX-loaded K specimens and 13.5 ± 0.58 KHN in thermally aged control PC specimens. Flexural strength medians varied from 42.0 ± 14.88 to 180.0 ± 57.31 MPa in chemically aged control K and thermally aged control PC specimens, respectively. The surface free energy changed between 24.0 ± 5.65 mM/m thermally aged control UFI specimens and 42.5 ± 0.28 mM/m chemically aged CHX-loaded UFI specimens.

After thermal and chemical aging, CHX-loaded K and UFI specimens did not reveal statistically significant (*p* > 0.05) differences in microhardness, flexural strength, and surface free energy compared to the control. However, the 5 wt% CHX-loaded PC specimens showed decreased microhardness (*p* = 0.010) and flexural strength (*p* = 0.038) and increased surface free energy (*p* = 0.008) after thermal aging, but only decreased (*p* = 0.021) flexural strength after chemical aging (Table 2).

The color change of the resins submitted to aging processes was also evaluated. Considering the thermal aging, the color change medians varied between 1.4 ± 1.62 ∆E for the CHX-loaded PC specimens and 10.3 ± 3.00 ∆E for the CHX-loaded K specimens. CHX-loaded K specimens showed statistically significant (*p* = 0.008) higher ∆E values than the control. On the other hand, CHX-loaded UFI and PC specimens did not show statistically significant (*p* = 0.548 and *p* = 1.000, respectively) differences in ∆E values than the control (Figure 1).

After chemical aging, the color change medians varied between 1.8 ± 3.36 ∆E in control PC specimens and 16.3 ± 2.13 ∆E in CHX-loaded PC specimens, revealing a statistically significant difference (*p* = 0.008). In addition, K and UFI acrylic resins with CHX showed higher ∆E values (*p* = 0.008) than controls (Figure 2).

Moreover, the data related to the NBS units and the color differences were determined (Table 3).

After thermal aging, CHX incorporation in K specimens reveals a “much appreciable change” in the color of the resin. This change was even more significant when the CHX-loaded K specimens “changed to another color” when submitted to chemical aging. Considering chemical aging, loading the acrylic resins PC and UFI with CHX caused a “change to another color” and an “appreciable change”, respectively.

## 4. Discussion

This study evaluated the behavior of three CHX-loaded reline polymeric resins used to treat denture stomatitis after being submitted to thermal and chemical aging, simulating oral biodegradation processes. The results showed that the direct K and UFI resins did not change their mechanical and surface properties when loaded with CHX but changed their color, especially after chemical aging. In addition, the indirect PC resin loaded with CHX reveals a decrease in mechanical properties after both aging processes and a color change when submitted to chemical aging.

The specimens were submitted to physical and chemical alterations in both aging processes to simulate 4–6 weeks of oral biodegradation. This time was selected since the authors concluded in a previous study that the highest CHX release from the studied resins occurred within the first 24–48 h of incubation, followed by a steadier and slower release rate for the next 2 weeks, and ending with a minor release from the 20th day until the end of testing (28 days) [23]. Presumably, since the resin does not leach more relevant CHX after this period, the properties of the bioactive resins after aging will be similar to the resin without incorporation. Further aging processes can add little information about the bioactive system degradation. In addition, this time selection is in line with other works that also choose 28 days as the threshold for testing bioactive systems [12,13,14].

Concerning thermal degradation, a laboratory testing procedure of 1000 cycles of thermal fluctuations between water baths of 5 °C and 55 °C, corresponding to 6 weeks in function, was used to simulate the temperature fluctuations that occur in the oral cavity when mouth breathing and intake of food and beverages with different temperatures [25,38,39]. As in the study of Bettencourt et al. [23] conducted without aging, the incorporation of CHX decreases the microhardness and flexural strength of PC specimens submitted to thermal aging. The decrease in these parameters may be due to CHX loading and further enhanced by high temperatures, which promote high leachability of CHX molecules from the material network to the surrounding medium, leaving spaces in the polymer net [40]. Finally, the water molecules penetrate the areas between the polymer chains and push them farther apart.

Consequently, the secondary chemical bonding forces (van der Waals forces) between the polymer chains decline [41,42]. As so, the water molecules can act as plasticizers that damage the material’s mechanical resistance through the formation of microcracks related to absorption and hydrolytic degradation of the polymer, resulting in linkage cleavage and gradual deterioration of its infrastructure [31]. On the contrary, the CHX incorporation did not influence the flexural strength and microhardness of K and UFI resins. Differences in the bulk structure of these direct reline materials can explain why the mechanical parameters were affected only for the indirect PC resin. Specifically, the PC resin has a more robust and rigid polymer matrix organization than K and UFI [43]. Thus, the flexural strength and microhardness reduction in the 5 wt% CHX-loaded PC group may be related to a higher increase in intermolecular distance between the polymer chains after the CHX leaching compared to the other two resins (K and UFI) [44].

Nevertheless, although PC showed a decrease in mechanical properties with the incorporation of CHX, under current experimental conditions, its median value was still higher than that obtained with K and UFI. In this situation, the 5 wt% CHX-loaded PC group still reached a flexural strength value clinically accepted by the ISO 20795 standard (60 MPa) [35,45,46]. Furthermore, the tested reline acrylic resins are not used independently in the oral cavity since they are always associated with a denture base material, so this result should be interpreted cautiously.

Along with thermal fluctuations, the media’s chemical environment was identified to influence the properties of polymer resins [30]. In the present study, specimens were subjected to a chemical aging process by immersion in artificial saliva with a cyclic procedure of 6 h at pH = 3, interchanging with 18 h at pH = 7 for 28 days. In the present study, only CHX-loaded PC specimens decreased flexural strength after chemical aging. Once again, this result shows that the inner structure of the CHX-loaded PC polymer suffers more changes by the aging processes, not only thermal but also chemical, compared to K and UFI resins.

Besides mechanical properties, the effect of aging on the reline resins’ surface properties was evaluated since surface energy plays an essential role in the wettability of the material which can interfere with the adhesion of fungal cells [33,47]. In the present study, only CHX-loaded PC specimens showed a minor increase in the surface free energy parameter after undergoing thermal aging than the control group. Thus, this indirect resin became slightly more hydrophilic (i.e., with high surface free energy) by increasing its polar component after the drug’s incorporation [40,42,48,49,50]. This tendency was also observed in previous studies without aging [23,47,50] and with other acrylic resins (Lucitone 550) loaded with an antimicrobial polymer poly (2-tert-butylaminoethyl) methacrylate [33].

The increase in surface energy can also be associated with changes in the surface layer due to the sorption of the surrounding fluids, particularly water, leading to an increase in the volume of the polymer [40,47]. Furthermore, the insignificant effect of aging on the surface energy of the K and UFI resins possibly suggests that these materials do not absorb saliva when subjected to chemical aging or change the structure of their surface layer after thermal aging keeping their dimensional over time [51].

The success of removable dentures depends not only on their mechanical and surface properties but also on their aesthetic appearance related to their color. For example, in acrylic reline resins, the mucosa-like pink color is obtained by adding pigments to the powder–liquid mixture during the polymerization.

The structure of the polymer is one factor that influences color stability since its intrinsic porosity favors the leachability of the pigment molecules and consequent replacement with molecules from the surrounding media, turning it sensitive to discoloration [52]. In the present study, this discoloration was shown since all the control specimens (without incorporation) underwent a color change after aging. In line with another study [8], our results showed that different resins presented distinct color changes. K showed an “appreciable change” of color (>3.3 NBS units as the limit for acceptability in clinical dentistry), while UFI and PC showed only perceivable change [33,53].

At last, the effect of the aging processes and CHX loading on the polymer´s color stability was evaluated. Considering the thermal aging, only K showed color differences when CXH was incorporated, going to a “much appreciable” change of color that is not considered clinically acceptable. As mentioned above, K comprises the isobutyl methacrylate monomer and forms a simple non-cross-linking PEMA net when the polymerization is complete [54]. However, since the polymer net is not robust, it is more accessible to the movements of internal pigments and water molecules, which may contribute to the formation of aberrant optical properties affecting the polymer´s color stability [30,55].

Additionally, all CHX-loaded specimens submitted to chemical aging showed a considered clinically relevant color change. For example, loading CHX to K caused a “change to another color”, and UFI caused an “appreciable change”. Moreover, PC resin showed a “change to another color”, probably related to its hydrophilic character, which was already shown to exhibit more significant color change than hydrophobic materials [56,57]. It should also be emphasized that chemical aging using an artificial saliva medium leads to a more evident color change than aging using just water [58,59,60]. These conclusions follow a previous study with soft lining materials [61,62], where the authors stated that a complex solution such as artificial saliva might affect the tested biomaterials more than plain water since saliva has more components, such as organic substances, including immunoglobulins, proteins, enzymes, and mucins [6,57].

In summary, there were clinically relevant color changes due to aging processes and CHX loading. However, considering that this bioactive system is used in reline resins placed under the denture resins, the impact on the color change may be insignificant to patients. Therefore, the benefit of a long-term therapeutic effect will likely outweigh the disadvantage of the lack of color stability.

Future investigations should be carried out by combining multiple aging methods (e.g., mechanical, thermal, and chemical) to better reproduce the oral environment. After the in vitro experiments, clinical studies should be designed and conducted to confirm the potential of this bioactive chlorhexidine system in treating denture stomatitis.

## 5. Conclusions

Within the limitations of this in vitro study, it can be established that CHX bioactive systems based on reline acrylic resins showed different results in mechanical and surface properties after aging. While direct reline resins did not show differences when CHX was incorporated, PC revealed decreased mechanical properties after thermal aging but not inadequate levels for function. In addition, chemical aging affects CHX bioactive systems since it presents a color change in all resins. Considering the overall results, loading CHX into reline acrylic resins do not impair the relevant mechanical and aesthetic functions of removable dentures and continues to be a potential approach in preventing or treating denture stomatitis, but further research is required before their possible clinical use.

## Figures and Tables

**Figure 1 polymers-15-02549-f001:**
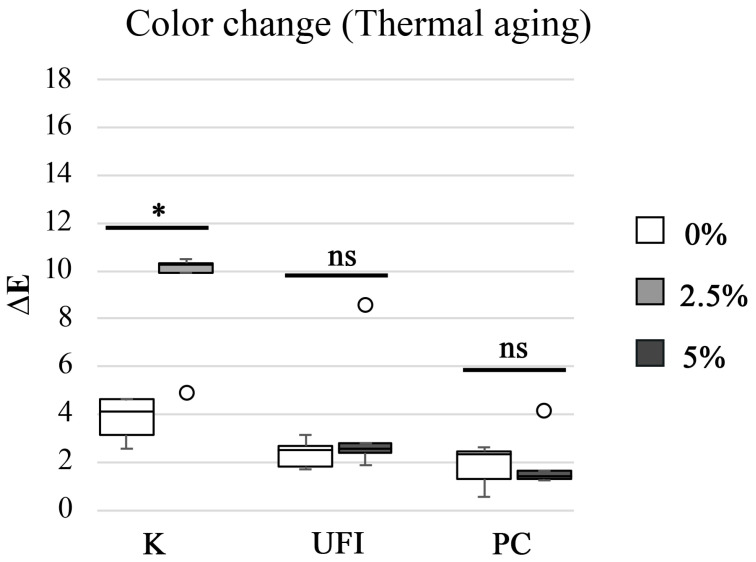
Boxplot of the color change (∆E) distribution among experimental groups after thermal aging. Note: * corresponds to *p* < 0.05, and ns corresponds to “not significant”.

**Figure 2 polymers-15-02549-f002:**
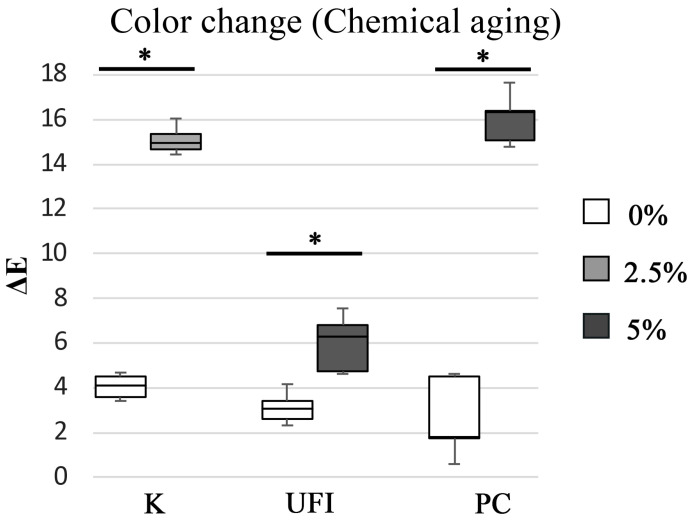
Boxplot of the color change (∆E) distribution among experimental groups after chemical aging. Note: * corresponds to *p*< 0.05.

**Table 1 polymers-15-02549-t001:** Dental materials under evaluation.

Material	Manufacturer	P/L Ratio (g/mL)	Composition	Curing Cycle
K	GC America Inc., Alsip, IL, USA	1.4/1	P: PEMA 88–90%, dibenzoyl peroxide 1-<2.5%, silicon and titanium dioxides 5-<10%, cellulose acetate <2.5% L: IBMA 90-<100%, N,N-dimethyl-p-toluidine 1-<2.5%	10 min 37 °C
UFI	VOCO GmbH, Cuxhaven, Germany	1.77/1	P: PEMA 90–95%, benzoyl peroxide ≤2.5% L: HDMA 50–100%, hydroxyethyl methacrylate ≤2.5%	7 min 37 °C
PC	Ivoclar Vivadent AG, Schaan, Liechtenstein	1.5/1	P: PMMA > 95%, softening agent <1%, benzoyl peroxide 1-<2.5%, catalyst, pigments L: MMA 50–100%, BDMA 2.5-<10%, catalyst	15 min 40 °C, 4 bar

K—Kooliner; UFI—Ufi Gel Hard; PC—Probase Cold; P—powder; L—liquid; PEMA—poly (ethyl methacrylate); IBMA—isobutyl methacrylate; HDMA—hexanediol dimethacrylate; PMMA—poly (methyl methacrylate); MMA—methyl methacrylate; BDMA—1,4-butanediol dimethacrylate.

**Table 2 polymers-15-02549-t002:** Microhardness (KHN), flexural strength (MPa), and surface free energy (γ) data by polymer resin.

RESIN-BASED POLYMER	CHX LOADING (wt%)	AGING PROCESS	MICROHARDNESS (KHN, kgf mm^−2^)		FLEXURAL STRENGTH (MPa)		SURFACE FREE ENERGY (mN/m)	
			MED ± IQR	*p*-Value	MED ± IQR	*p*-Value	MED ± IQR	*p*-Value
K	0	Thermal	7.9 ± 2.73	*p* = 1.000	78.9 ± 26.00	*p* = 0.054	27.9 ± 4.45	*p* = 0.222
	2.5		7.9 ± 1.60		91.1 ± 17.63		27.0 ± 1.85	
	0	Chemical	7.2 ± 2.97	*p* = 0.798	42.0 ± 14.88	*p* = 0.959	31.8 ± 2.95	*p* = 0.222
	2.5		7.0 ± 3.44		42.1 ± 12.89		33.4 ± 3.05	
UFI	0	Thermal	8.2 ± 1.85	*p* = 0.878	67.0 ± 20.08	*p* = 0.130	24.0 ± 5.65	*p* = 0.095
	5		8.4 ± 0.54		75.6 ± 13.60		32.0 ± 6.50	
	0	Chemical	7.6 ± 2.25	*p* = 0.878	36.5 ± 6.51	*p* = 0.645	41.8 ± 2.70	*p* = 0.548
	5		7.9 ± 1.94		37.4 ± 5.60		42.5 ± 0.28	
PC	0	Thermal	13.5 ± 0.58	*p* = 0.010	180.0 ± 57.31	*p* = 0.038	26.3 ± 0.40	*p* = 0.008
	5		12.8 ± 0.53		124.6 ± 38.75		30.5 ± 2.40	
	0	Chemical	13.1 ± 4.17	*p* = 0.195	87.3 ± 19.04	*p* = 0.021	37.2 ± 3.30	*p* = 0.841
	5		12.4 ± 3.94		65.6 ± 9.74		36.6 ± 2.25	

K—Kooliner; UFI—Ufi Gel Hard; PC—Probase Cold; CHX—chlorhexidine; MED—median; IQR—interquartile range.

**Table 3 polymers-15-02549-t003:** NBS units and color differences by polymer resin.

POLYMER RESIN	CHX LOADING wt%	AGING PROCESS	M ± SD NBS Units	Color Differences
K	0	Thermal	3.5 ± 0.85	Appreciable change
	2.5		8.5 ± 2.21	Much appreciable
	0	Chemical	3.7 ± 0.50	Appreciable change
	2.5		13.9 ± 0.58	Change to another color
UFI	0	Thermal	2.2 ± 0.56	Perceivable change
	5		3.4 ± 2.58	Appreciable change
	0	Chemical	2.9 ± 0.66	Perceivable change
	5		5.5 ± 1.19	Appreciable change
PC	0	Thermal	1.7 ± 0.82	Perceivable change
	5		1.8 ± 1.15	Perceivable change
	0	Chemical	2.4 ± 1.65	Perceivable change
	5		14.8 ± 1.08	Change to another color

CHX—chlorhexidine; M—medium; SD—standard deviation.

## Data Availability

Data will be available on request.

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
