# Peer review of "Understanding the Mechanical, Surface, and Color Behavior of Oral Bioactive Prosthetic Polymers under Biodegradation Processes"

_polymers, 2023, doi:10.3390/polym15112549_

Round 1

Reviewer 1 Report

The present article deals with the micromechanical, macromechanical, and esthetic properties of denture reline resins doped with chlorhexidine to make them antibacterial. The study has a clear and well-developed rationale and evaluates important aspects of preclinical testing in an attempt to reduce the incidence of denture stomatitis. The methodology is appropriate for the objectives of the study, and the results are well described and elaborated. Overall, the manuscript answers the research question well. However, I would like to make some (non-critical) comments for the authors to consider.

1. Why is thermocycling used for only 1000 cycles (corresponding to 6 weeks of clinical use)? Should not the relined prosthesis be functional for much longer? Why is not simulating e.g. one year of clinical service? Presumably, further degradation should occur after this time.

2. Why is not the flexural modulus evaluated in addition to the flexural strength? This might give you better insight into material degradation. By performing the three-point bending test, you have already collected all the data needed to calculate the flexural modulus. So no further experiments are needed, just some simple calculations. Please consider whether it may be beneficial to add the data on the flexural modulus.

3. In the Discussion section, I do not quite understand what is meant by the word "inner" in the sentence "Differences in the inner structure of these direct reline materials can explain why the mechanical parameters were affected only for the indirect PC resin." - Did you mean to say "properties of the bulk material", as opposed to surface properties? Please clarify.

4. Another language issue: in the Conclusion section, in the sentence "Considering the overall results, loading CHX into reline acrylic resins do not impair the proper mechanical and aesthetic functions of removable dentures", maybe the word "proper" should be replaced with "relevant".

The English is fine, except for some minor problems with the choice of words (see above)

Author Response

The present article deals with the micromechanical, macromechanical, and esthetic properties of denture reline resins doped with chlorhexidine to make them antibacterial. The study has a clear and well-developed rationale and evaluates important aspects of preclinical testing in an attempt to reduce the incidence of denture stomatitis. The methodology is appropriate for the objectives of the study, and the results are well described and elaborated. Overall, the manuscript answers the research question well. However, I would like to make some (non-critical) comments for the authors to consider.

Q1. Why is thermocycling used for only 1000 cycles (corresponding to 6 weeks of clinical use)? Should not the relined prosthesis be functional for much longer? Why is not simulating e.g. one year of clinical service? Presumably, further degradation should occur after this time.

Answer: The release assays performed in previous work [23] showed that the highest CHX release occurred within the first 24–48 h of incubation, indicating a surface release process. Then, a steadier and slower release rate for the next two weeks caused by the drug diffusion from the bulk of the polymer was observed, followed by a minor release from the 20th day until the end of testing (28 days). Presumably, since the resin does not leach more relevant CHX after this period, the mechanical behavior of the bioactive resin after 4-6 weeks will be similar to the resin without incorporation. Further aging processes can add little information about the bioactive system degradation.

The authors appreciate this comment, and a paragraph referring to this issue was added to the Discussion chapter “The specimens were submitted to physical and chemical alterations in both aging processes to simulate 4-6 weeks of oral biodegradation. This time was selected since the authors concluded in a previous study that the highest CHX release from the studied resins occurred within the first 24–48 h of incubation, followed by a steadier and slower release rate for the next two weeks, and ending with a minor release from the 20th day until the end of testing (28 days) [23]. Presumably, since the resin does not leach more relevant CHX after this period, the properties of the bioactive resins after aging will be similar to the resin without incorporation. Further aging processes can add little information about the bioactive system degradation. Also, this time selection is in line with other works that also choose 28 days as the threshold for testing bioactive systems [12-14]”.

Q2. Why is not the flexural modulus evaluated in addition to the flexural strength? This might give you better insight into material degradation. By performing the three-point bending test, you have already collected all the data needed to calculate the flexural modulus. So no further experiments are needed, just some simple calculations. Please consider whether it may be beneficial to add the data on the flexural modulus.

 Answer: The flexural modulus of a material measures the stiffness or resistance to bending when a force is applied to it. A calculation of this mechanical property includes the deflection (displacement in the material) that depends on material composition, temperature, strain rate, and manufacturing process. Unfortunately, the authors can no longer access that measurement calculated during testing. The authors appreciate this pertinent suggestion and include flexural modulus in further studies.

Q3.In the Discussion section, I do not quite understand what is meant by the word "inner" in the sentence "Differences in the inner structure of these direct reline materials can explain why the mechanical parameters were affected only for the indirect PC resin." - Did you mean to say "properties of the bulk material", as opposed to surface properties? Please clarify.

Answer: The authors are thankful for this comment and changed the word “inner” for “bulk” in the Discussion section.

Q4.Another language issue: in the Conclusion section, in the sentence "Considering the overall results, loading CHX into reline acrylic resins do not impair the proper mechanical and aesthetic functions of removable dentures", maybe the word "proper" should be replaced with "relevant".

Answer: The authors appreciated the suggestion and corrected the Conclusion section, changing the word “proper” to “relevant”.

Reviewer 2 Report

The authors presented an interesting paper in which the problem of chemical interaction and conditions of the oral cavity on the functional properties of oral bioactive prosthetic polymers was addressed. The main purpose of this work was to evaluate the mechanical, surface, and color of CHX-loaded polymeric resins after undergoing physical or chemical aging that mimics 1-month of exposition in the oral cavity. The scope and plan of research is appropriate to the assumed purpose of the work. The applied conditioning, surface properties, bulk strength testing and testing of color differences procedures are appropriate for the test items. The results are properly compiled and statistical analyzes are sufficient to assess the differences between the experimental groups.

Below are my comments:

Table 1 does not specify the weight/volume content of the components of the tested materials.

For material K, 2.5 wt.% CHX was used. For UFI and PC materials 5 wt.% CHX. Reference is made to work [28]. This is an important point in the article. Let me explain this difference briefly.

L 172: Invalid footnote.

L 167: I propose to specify the load in the microhardness test as 0.1 kgf or 10 gf in brackets next to the value in mN. The kgf/gf unit is more familiar and more commonly used among researchers.

In Table 2, the Knoop hardness unit can be given as kgf·mm−2

L 236: The name of the National Bureau of Standards (NBS) is given. Shouldn't this name be different? Maybe it should be "National Institute of Standards and Technology (NIST)"? Please check.

L 257: Invalid footnote.

In the discussion chapter, the authors focused on the practical and clinical significance of the research results. What is essential. However, in my opinion, the research results do not relate to the expected time of use of the bioactive chlorhexidine system in the treatment of patients in clinical conditions. To what extent do 1000 hydro-thermal cycles and 28 days in artificial saliva correspond to the clinical conditions of the tested bioactive system? This thread should be developed in the discussion. The authors do not devote much space in the discussion to explaining the mechanisms of structural changes in the surface layer and the volume of polymer material samples. What did you expect from the job title? Appropriate only when reference is made to [45] is this mentioned. I propose to add a paragraph in which the description of degradation of the surface (top layer) and volume of polymeric materials due to conditioning can be developed.

Author Response

The authors presented an interesting paper in which the problem of chemical interaction and conditions of the oral cavity on the functional properties of oral bioactive prosthetic polymers was addressed. The main purpose of this work was to evaluate the mechanical, surface, and color of CHX-loaded polymeric resins after undergoing physical or chemical aging that mimics 1-month of exposition in the oral cavity. The scope and plan of research is appropriate to the assumed purpose of the work. The applied conditioning, surface properties, bulk strength testing and testing of color differences procedures are appropriate for the test items. The results are properly compiled and statistical analyzes are sufficient to assess the differences between the experimental groups.

Below are my comments:

Q1. Table 1 does not specify the weight/volume content of the components of the tested materials.

Answer: The authors appreciated the suggestion and added information on the components of the tested materials as detailed by the manufacturers.

Q2. For material K, 2.5 wt.% CHX was used. For UFI and PC materials 5 wt.% CHX. Reference is made to work [28]. This is an important point in the article. Let me explain this difference briefly.

Answer: The authors thank the reviewer for this suggestion. Antimicrobial testing results from a previous study [23] showed that the lowest CHX concentration promoting antimicrobial activity against C. albicans was 2.5 wt% for K and 5 wt% for UFI and PC. Differences in the chemical composition and physical structure of the resins can explain the difference since the leaching of CHX can be affected by the organization of the polymer matrix, level of porosity, and water sorption.

The authors added some brief explanation in the Material and Methods section “The selection of the CHX concentrations was based on the results of a previous study corresponding to the minimal concentration that promoted antimicrobial activity against C. albicans in each structurally distinct resin”.

Q3. L 172: Invalid footnote.

Answer: The correction was made (ISO was replaced by reference 35).

Q4: L 167: I propose to specify the load in the microhardness test as 0.1 kgf or 10 gf in brackets next to the value in mN. The kgf/gf unit is more familiar and more commonly used among researchers.

Answer. As the peer reviewer suggested, the load in microhardness is also specified in gf.

Q5: In Table 2, the Knoop hardness unit can be given as kgf·mm−2

Answer. As the peer reviewer suggested, in Table 2, the authors added the unit kgf·mm−2 to Knoop hardness.

Q6: L 236: The name of the National Bureau of Standards (NBS) is given. Shouldn't this name be different? Maybe it should be "National Institute of Standards and Technology (NIST)"? Please check.
Answer. Although the National Bureau of Standards was renamed the National Institute of Standards and Technology in 1988, the scientific literature still uses the NBS units to reference critical remarks on color differences, as recent examples are given below (1-3). To permit further comparison with other works, the authors decided to maintain the identification in NSB units.

1- Hotta M, Murase Y, Shimizu S, Kusakabe S, Takagaki T, Nikaido T. Color changes in bulk-fill resin composites as a result of visible light-curing. Dent Mater J. 2022 Feb 1;41(1):11-16. doi: 10.4012/dmj.2021-032.

2- Alfouzan AF, Alotiabi HM, Labban N, Al-Otaibi HN, Al Taweel SM, AlShehri HA. Color stability of 3D-printed denture resins: effect of aging, mechanical brushing and immersion in staining medium. J Adv Prosthodont. 2021 Jun;13(3):160-171. doi: 10.4047/jap.2021.13.3.160.

3- Azmy E, Al-Kholy MRZ, Gad MM, Al-Thobity AM, Emam AM, Helal MA. Influence of Different Beverages on the Color Stability of Nanocomposite Denture Base Materials. Int J Dent. 2021 Nov 11;2021:5861848. doi: 10.1155/2021/5861848.

Q7: 257: Invalid footnote.

Answer: The reviewer is correct; the correction was done.

Q8. In the discussion chapter, the authors focused on the practical and clinical significance of the research results. What is essential. However, in my opinion, the research results do not relate to the expected time of use of the bioactive chlorhexidine system in the treatment of patients in clinical conditions. To what extent do 1000 hydro-thermal cycles and 28 days in artificial saliva correspond to the clinical conditions of the tested bioactive system? This thread should be developed in the discussion. 

Answer: The release assays performed in previous work [23] showed that the highest CHX release occurred within the first 24–48 h of incubation, indicating a surface release process. Then, a steadier and slower release rate for the next two weeks caused by the drug diffusion from the bulk of the polymer was observed, followed by a minor release from the 20th day until the end of testing (28 days). Presumably, since the resin does not leach more relevant CHX after this period, the mechanical behavior of the bioactive resin after 4-6 weeks will be similar to the resin without incorporation. Further aging processes can add little information about the bioactive system degradation.

The authors appreciate this comment, and a paragraph referring to this issue was added to the Discussion chapter “The specimens were submitted to physical and chemical alterations in both aging processes to simulate 4-6 weeks of oral biodegradation. This time was selected since the authors concluded in a previous study that the highest CHX release from the studied resins occurred within the first 24–48 h of incubation, followed by a steadier and slower release rate for the next two weeks, and ending with a minor release from the 20th day until the end of testing (28 days) [23]. Presumably, since the resin does not leach more relevant CHX after this period, the properties of the bioactive resins after aging will be similar to the resin without incorporation. Further aging processes can add little information about the bioactive system degradation. Also, this time selection is in line with other works that also choose 28 days as the threshold for testing bioactive systems [12-14].”

Q9. The authors do not devote much space in the discussion to explaining the mechanisms of structural changes in the surface layer and the volume of polymer material samples. What did you expect from the job title? Appropriate only when reference is made to [45] is this mentioned. I propose to add a paragraph in which the description of degradation of the surface (top layer) and volume of polymeric materials due to conditioning can be developed.

Answer: In line with the reviewer 's suggestion, the authors have improved the Discussion section, and a detailed comparison of surface energy results with other studies, including more recent ones, was included. Also, a paragraph was added to discuss the possible negligible effect of aging on the surface (top layer) and volume of the resins: “The increase in surface energy can also be associated with changes in the surface layer due to the sorption of the surrounding fluids, particularly water, leading to an increase in the volume of the polymer [40,47]. Furthermore, the insignificant effect of aging on the surface energy of the K and UFI resins possibly suggests that these materials do not absorb saliva when subjected to chemical aging or change the structure of their surface layer after thermal aging keeping their dimensional over time [51]”.

Reviewer 3 Report

I think this manuscript could be accepted after minor revision.

The authors should be asked to address the following queries. 1. The introduction part should be revised. It should be more condensed, with clearer transitions between themes. The novelty of the work should be emphasized discussing current literature. 2. Figure numbers should be checked. The numbering is incomplete and the same numbers are written. 3. Discussion parts are so poor and authors should compare their results with recent studies (2015-2023).   

Author Response

The authors should be asked to address the following queries.

Q1. The introduction part should be revised. It should be more condensed, with clearer transitions between themes. The novelty of the work should be emphasized discussing current literature. 

Answer: The introduction was revised and condensed with more explicit transitions between the themes. The work's novelty was further emphasized compared to the current literature.

Q2. Figure numbers should be checked. The numbering is incomplete and the same numbers are written. 

Answer: The figures were checked and corrected.

Q3. Discussion parts are so poor and authors should compare their results with recent studies (2015-2023).   

Answer: The authors acknowledge the reviewer's comment. In line with the reviewer's suggestion, the Discussion section was significantly improved:

- A detailed comparison of the surface energy results with other studies, including more recent ones, was included. Also, a paragraph was added to discuss the possible negligible effect of aging on the surface (top layer) and volume of the resins: “The increase in surface energy can also be associated with changes in the surface layer due to the sorption of the surrounding fluids, particularly water, leading to an increase in the volume of the polymer [40, 47]. Furthermore, the insignificant effect of aging on the surface energy of the K and UFI resins possibly suggests that these materials do not absorb saliva when subjected to chemical aging or change the structure of their surface layer after thermal aging keeping their dimensional over time [51] “.

-  An explanation of the biodegradation simulation time of the tested bioactive systems through the thermal and chemical aging process was developed in a paragraph: “The specimens were submitted to physical and chemical alterations in both aging processes to simulate 4-6 weeks of oral biodegradation. This time was selected since the authors concluded in a previous study that the highest CHX release from the studied resins occurred within the first 24–48 h of incubation, followed by a steadier and slower release rate for the next two weeks, and ending with a minor release from the 20th day until the end of testing (28 days) [23]. Presumably, since the resin does not leach more relevant CHX after this period, the properties of the bioactive resins after aging will be similar to the resin without incorporation. Further aging processes can add little information about the bioactive system degradation. Also, this time selection is in line with other works that also choose 28 days as the threshold for testing bioactive systems [12-14].”